# Simple Self-Assembly Strategy of Nanospheres on 3D Substrate and Its Application for Enhanced Textured Silicon Solar Cell

**DOI:** 10.3390/nano11102581

**Published:** 2021-09-30

**Authors:** Dan Su, Lei Lv, Yi Yang, Huan-Li Zhou, Sami Iqbal, Tong Zhang

**Affiliations:** 1Key Laboratory of Micro-Inertial Instrument and Advanced Navigation Technology, Ministry of Education, School of Instrument Science and Engineering, Southeast University, Nanjing 210096, China; jssysls@163.com; 2Joint International Research Laboratory of Information Display and Visualization, School of Electronic Science and Engineering, Southeast University, Nanjing 210096, China; lvlei@seu.edu.cn (L.L.); yyang19x@163.com (Y.Y.); huanli_zhou@163.com (H.-L.Z.); 101300112@seu.edu.cn (S.I.); 3Suzhou Key Laboratory of Metal Nano-Optoelectronic Technology, Southeast University Suzhou Campus, Suzhou 215123, China

**Keywords:** self-assembly, 3D substrate, solar cell

## Abstract

Nanomaterials and nanostructures provide new opportunities to achieve high-performance optical and optoelectronic devices. Three-dimensional (3D) surfaces commonly exist in those devices (such as light-trapping structures or intrinsic grains), and here, we propose requests for nanoscale control over nanostructures on 3D substrates. In this paper, a simple self-assembly strategy of nanospheres for 3D substrates is demonstrated, featuring controllable density (from sparse to close-packed) and controllable layer (from a monolayer to multi-layers). Taking the assembly of wavelength-scale SiO_2_ nanospheres as an example, it has been found that textured 3D substrate promotes close-packed SiO_2_ spheres compared to the planar substrate. Distribution density and layers of SiO_2_ coating can be well controlled by tuning the assembly time and repeating the assembly process. With such a versatile strategy, the enhancement effects of SiO_2_ coating on textured silicon solar cells were systematically examined by varying assembly conditions. It was found that the close-packed SiO_2_ monolayer yielded a maximum relative efficiency enhancement of 9.35%. Combining simulation and macro/micro optical measurements, we attributed the enhancement to the nanosphere-induced concentration and anti-reflection of incident light. The proposed self-assembly strategy provides a facile and cost-effective approach for engineering nanomaterials at 3D interfaces.

## 1. Introduction

The development of nanomaterials and nanostructures provides new opportunities for performance boosting of optical and optoelectronic devices [1,2,3,4,5]. For example, dielectric nanostructures are used to enhance the transmission and brightness of different transparent windows or display screens [6,7,8,9], and plasmonic or dielectric nanocoatings are widely proposed and applied to enhance the efficiency of photovoltaic devices or sensitivity of photodetectors [1,10]. In many optoelectronic devices, there are complex 3D surfaces [11,12]. For example, micron and nanoscale heterogeneities exist in various photovoltaic devices [13], such as the intrinsic grains in polycrystalline solar cells [14,15,16] and textured surfaces in the commercial silicon solar cells for light-trapping [17].

In this context, there has been a drive for designing and fabricating nanostructures on 3D surfaces with desired control. On planar substrates or interfaces, both top-down and bottom-up self-assembly strategies enable precise control over distribution density and geometrical shapes [8,18,19,20,21,22,23]. Although substantial advances have been made, control over nanofabrication on complex 3D surfaces is still very challenging [7]. Top-down strategies, such as electron beam lithography and focused ion beam, show some difficulties in the fabrication on 3D substrates limited by the inherent process characteristics of photoresist coating or beam focus [24]. Self-assembly methods, using colloidal nanoparticles as building blocks, show great potential in assembling nanostructures on 3D substrates [18]. Previous reports have demonstrated the self-assembly results on 3D fabrics and textured wafers [25,26,27]. However, the nanostructures usually accumulate and form multi-layers at the bottom of valleys [25,26], while precise control over the distribution density and layer numbers on the 3D substrate has remained elusive.

We emphasize the importance of controllable nanosphere assembly for understanding the mechanism of nanosphere-enhanced textured photovoltaics. For photovoltaic devices application, wavelength scale dielectric nanospheres have attracted much attention. Nanosphere array supports Whispering Gallery Modes (WGM), which can be coupled with the guided modes in photovoltaic thin-film to improve the light absorption of the devices [10,28]. Some studies have shown that dielectric microspheres can also enhance planar bulk material cells [29,30], mainly due to the anti-reflection effect of close-packed SiO_2_ spheres. Besides, SiO_2_ arrays enable colorful photovoltaic devices without sacrificing device performance [31]. In 2020, Bek et al. proposed that in planar cells, coexisting anti-reflection and light concentration effects are the reason for the nanosphere enhancement [32]. The former contributes to an enhanced photocurrent, and the latter contributes to an enhanced fill factor of the devices. When it comes to textured solar cells, it is highly desirable to control nanosphere distribution density with specific layer numbers to explore the optimized parameters and discern the underlying mechanism.

In this work, we propose a simple assembly strategy on 3D substrates featuring controllable distribution density and coating layers. The influence of gravitational sedimentation on assembly behavior was analyzed by comparing substrates with different surface features (planar and textured) and under different orientations (upward and inverted) in the SiO_2_ colloid. The effect of SiO_2_ microsphere coating on the device performance under different assembly conditions was further explored. For the first time, the nanosphere anti-reflection and light concentration mechanisms were analyzed on the textured solar cells with numerical simulation, macro and micro-region optical characterization.

## 2. Materials and Methods

### 2.1. Materials

Tetraethyl orthosilicate (TEOS, 98%) and ammonium hydroxide solution (NH_4_OH, 28.0–30.0%) were bought from Sigma-Aldrich (Saint Louis, MO, USA). Absolute ethanol (≥95%) was purchased from Sinopharm Chemical Reagent Co., Ltd. (Ningbo, China). Poly-l-lysine solution has a concentration of 0.1 mg/mL and degree of polymerization from 144 to 335, purchased from Xiya Regent (Shanghai, China). Ultrapure water used in the experiments has a resistivity larger than 18.2 MΩ·cm (Milli-Q). All chemicals were used without any purification. Shanghai Shenzhou New Energy Co., Ltd. (Shanghai, China) supplied six-inch polycrystalline silicon solar cells. The solar cells were cut into 4 cm^2^ ones.

### 2.2. Synthesis of SiO_2_ Nanosphere

The ~550 nm diameter SiO_2_ nanosphere was synthesized according to previous work [33]. Briefly, 25 mL ultrapure water, 62 mL ethanol, and 9 mL NH_4_OH were mixed and stirred at a moderate speed for 30 min. Then, 4.5 mL TEOS was added into the mixed solution and reacted for 3 h at 30 °C. The resulting products were centrifuged three times using absolute ethanol and dried in an oven at 200 °C for an hour to obtain SiO_2_ nanosphere powders for future use. For the sake of a fast self-assembly process, SiO_2_ nanospheres were redispersed in absolute ethanol with a concentration of 50 mg/mL, and we call it SiO_2_ assembly solution.

### 2.3. Self-Assembly Process of SiO_2_ Nanosphere

The four-step self-assembly processes are outlined schematically in Figure 1. We first immersed the substrate into the poly-l-lysine solution (pH value about 6.2) for 4 min. The ionic form of the ε-amino group in poly-l-lysine solution depends primarily on the pH [34,35]. In the acid environment, the ε-amino group is in a positive charge state (–NH_3_^+^), widely used in biology and chemistry [34,36]. In such a dip process, the absorption of poly-L-lysine on the textured silicon substrate yields a positively charged surface. Then we used ultra-pure water to remove the redundant poly-l-lysine. Following this, the substrates were dip-coated onto the solution of SiO_2_ assembly solution for different time durations. It has been reported that the solution synthesized SiO_2_ nanospheres with an isoelectric point (IEP = 2.0) [37]. At a pH value higher than 2, the SiO_2_ nanospheres are abundant with OH^−^ and negatively charged. The negatively charged SiO_2_ surface will promote the electrostatic attraction to the positive textured surface while resisting the absorption of multi-layer SiO_2_. At last, the silicon solar cell was rinsed with ultra-pure water a second time to remove the unabsorbed SiO_2_ particles, which are not adhesive to the devices. It should be noticed that if the first two steps are not adopted, the SiO_2_ nanospheres could not be maintained on the surface after washing with ultra-pure water.

When the assembly process is repeated, another layer of poly-l-lysine will be absorbed upon the top surface of the SiO_2_ nanocoating assembled on the textured surface, forming a positive surface and enabling a layer-by-layer assembly process.

### 2.4. Characterization of Textured Silicon Solar Cells

Brightfield reflection and darkfield scattering optical images are taken from a Nikon (Ti-U) inverted microscope under the illumination of a halogen lamp. The grayscale transformation and statical analysis of the optical micro-region images were conducted using a MATLAB code. A long working distance 100×/0.80 NA objective lens (Nikon Plan Fluor ELWD 100×) was used to collect bright and darkfield images. The reflection spectra were taken from a fiber spectrometer (Nova, Ideaoptics Co., Ltd., Shanghai, China) coupled with an integrating sphere (IS-30-6-R). Electrical properties of solar cells are measured with a high-precision source-meter-unit (SMU, Keithley 2651) under the illumination of a 1000 W xenon lamp equipped with an AM1.5 filter (Crowntech Inc., Indianapolis, IN, USA).

### 2.5. Modeling of Silicon Solar Cell with SiO_2_ Nanospheres

The three-dimensional finite element method (COMSOL software) was used to simulate the light-concentrating performance of silica nanospheres, and the simulation process is similar to previously reported work [32]. In our simulation model, a single silica nanosphere or a vertically aligned dimmer is placed on the surface of the silicon substrate. The simulation width of the device is 2000 nm, the size of the nanoparticle is 550 nm, and the thickness of the air layer is 1500 nm. The linear-polarization plane wave is used as the incident excitation, and its amplitude is 1 V/m. In addition, to eliminate the unwanted reflection of the interface, the boundary of the device region was selected and was delimited by the perfectly matched layer (PML). The optical constants of SiO_2_ and Si are obtained by linear interpolation in the optical manual [38].

## 3. Results and Discussion

### 3.1. Self-Assembled SiO_2_ Nanocoating on Planar and Textured 3D Substrates

In order to fully demonstrate the assembly characteristics of the SiO_2_ nanosphere coating on the 3D textured surfaces, we compared the assembly results on the flat surface and the texturing Si wafer surface under the same assembly time. As shown in Figure 2a, on the surface of the flat substrate, the assembly of SiO_2_ nanospheres has two characteristics. Firstly, some of the SiO_2_ nanospheres become aggregated during the assembly process. Similar aggregates, also found in previously reported gold or dielectric nanocoatings via electrostatic assembly [39,40], can be attributed to the capillary force [41] (originating from water bridges formed between particles when the substrate is taken out of the solution). Secondly, the covered area of nanospheres is estimated at less than 35% according to SEM analysis (using ImageJ software packages). This situation has also been widely studied in previous electrostatic assemblies [39,42]. In the absence of external force, the assembly process of spherical nanoparticles usually conforms to the random sequential adsorption (RSA) model [43], where particles are supposed to be fixed on the substrate and cannot be moved after adsorption. Under this model, the surface coverage of particles could not exceed the jamming limit (54.7%) [39]. Furthermore, the repulsion between charged particles will further reduce surface coverage. Therefore, in the self-assembly coating on a planar substrate without external forces, the surface coverage of nanospheres usually does not exceed 40% [39].

Under the same assembly time, we observed that the particle coverage of silica nanospheres on the 3D substrate is significantly larger than that of a two-dimensional substrate, as shown in Figure 2b. Except for the sharp features, a single layer of densely arranged SiO_2_ nanospheres is almost formed on the entire substrate. Note that this assembly result is significantly different from the multi-layer accumulation only located at the bottom of the valley obtained by spin-coating or dip-coating [25,26]. In addition, our dense arrangement result is similar to the coating effect formed by the Langmuir-budget liquid-liquid assembly with an external force and the following transfer method [31]. However, our assembly method is simple and does not require the introduction of other external forces. We infer that the dense nanosphere assembly originates due to gravitational sedimentation [44]. For ~550 nm particles, the gravitational potential energy of the particle is comparable to the thermal energy accounting for thermal motion [45]. Therefore, the sedimentation of nanospheres in micron-scale valleys may lead to dense assembly behavior. Previously, the literature also proposed that even if there is repulsion between particles, nanoparticles (100–1000 nm) will still undergo sedimentation in the solvent [45].

To verify the influence of gravitational sedimentation in the assembly process, the textured substrates were set at different orientations in the SiO_2_ colloid. As shown in Figure 3, the assembly procedures with the substrate upward (Figure 3a) and inverted (Figure 3d) were performed under the same assembly time. The resultant assembly characteristics under the different orientations of substrate changed dramatically. A relatively dense assembly is achieved by an upward substrate (shown in Figure 3b,c). However, when the substrate was placed invertedly in the SiO_2_ colloid, the resultant surface coverage of SiO_2_ nanospheres was obviously reduced (shown in Figure 3d,e). The assembly time was increased to 30 min for further investigation of the assembly characteristics. As shown in Appendix A, for substrates with different assembly orientations, the surface coverages of SiO_2_ nanospheres were similar compared to that of 10 min assembly time. It can be concluded that the gravitational sedimentation contributes to a dense-packed nanocoating in the textured solar cell when the substrate is placed upward. Notably, gravitational sedimentation is a natural phenomenon, and therefore the dense-packed assembly can be achieved without other external forces such as electric field force or pressure [8,19,31].

The assembly behavior of nanoparticles on 3D texturing substrates (textured Si solar cells) under different timelines (10 s, 30 s, 10 min) and different assembly rounds was further investigated. The samples for different assembly times were named sample 1 (S1), sample 2 (S2), and sample 3 (S3) for 10 s, 30 s, 10 min, respectively. Figure 4a–i is one round of self-assembly SEM images and schematic diagrams of different assembly times. It can be found that, in the initial stage of assembly (10 s), the particles first deposit at the valley bottom, which indicates that the gravity of the nanoparticles is greater than the particle-substrate electrostatic attraction in this stage. As the assembly process proceeds, the silica nanospheres gradually assemble along the sidewalls of the valley. Subsequently, we repeated the assembling process twice, and the SEM images are shown in Figure 4j,k. As the schematic diagram Figure 4l shows, multiple layers appeared on the surface of the textured substrate, especially inside the valley. The sample for two round assembly was named sample 4 (S4).

### 3.2. The Performance Analysis of Textured Si Solar Cells with SiO_2_ Nanosphere Coatings

We compared the effects of different nanosphere coatings (different distribution densities and number of layers) on the electrical performance of the textured Si solar cells to obtain optimized parameters, as shown in Table 1 and Figure 5. Figure 5a shows a structural diagram of the textured Si solar cell with nanocoatings. The polycrystalline silicon solar cell thickness is ~180 μm, and the junction depth is about 500 nm. The SiO_2_ nanosphere coating is assembled on the textured upper surface of the device. As shown in Table 1, we tested the electrical properties of the devices under a solar simulator with an irradiance of 1000 W/m^2^. In the case of a single-layer SiO_2_ coating, as the distribution density increases, the relative enhancement of device efficiency gradually increases. When the nanospheres are in close packing, the maximum efficiency enhancement reaches 9.35%.

With a two-layer SiO_2_ coating (S4), the improvement is lower than that of the single-layer close-packed situation. So, we achieve an optimized self-assembly distribution for SiO_2_ coating on textured Si solar cells. Further optimizations in particle size or dielectric constant of nanospheres may obtain a higher efficiency enhancement, as discussed in planar devices [10].

Further, we hope to discuss the possible enhancement mechanism via analyzing the changing trends of electrical properties for different nanocoating, as shown in Figure 5b,c. We first noticed that the relative enhancement of short-circuit current density (*J*_sc_) follows the trend of enhancement of the solar cell efficiency, and the maximum relative enhancement is achieved under the single-layer close-packed situation. More importantly, it is noteworthy that efficiency enhancement is more significant than the *J*_sc_ enhancement. The results are different from the previous results on thin-film solar cells, where the efficiency enhancements were almost the same compared to *J*_sc_ enhancement [28]. Therefore, we further compared the open-circuit voltage (*V*_oc_) and fill factor of the devices, shown in Figure 5c. It is also found that the *V*_oc_ has a slight increase after coating with SiO_2_ nanospheres due to the logarithmic relation between *V*_oc_ and *J*_sc_.

Next, we focused on the interesting changing trend of fill factor after adding nanosphere coatings. When the density increased under single-layer conditions (S1, S2, and S3), the fill factor of the device gradually increased, which is similar to the changing trend of the current. However, although the *J*_sc_ enhancement of S4 (with two layers of SiO_2_ coating) is very close to that of S2 (single-layer, non-close-packed), their fill factor enhancement is significantly different (0.46% compared to 1.79%). This phenomenon indicates that multiple mechanisms exist accounting for enhancing the *J*_sc_ and fill factor of the textured solar cells. In the applications of dielectric nanospheres for enhanced photovoltaics, *J*_sc_ enhancement relates to more light being coupled into the photovoltaic device, increasing the generation rate of the photo-generated carriers [28,29,30]. Moreover, the increase in the fill factor was recently discovered and elucidated in planar solar cells [32]. It was attributed to the nanosphere concentrating effects. Here, we also found direct evidence for fill factor enhancement on textured solar cells, indicating the nanosphere’s light concentration could also enhance the efficiency of textured solar cells. Our results suggest that the nanosphere’s concentrating effect enables further efficiency enhancement, even under similar light absorption enhancement (with similar *J*_sc_ enhancement).

In order to clarify the light concentration effects for S2 and S4, we simplify the model to a single nanostructure (a single SiO_2_ sphere or two vertically arranged SiO_2_ dimmers) on a flat Si substrate. In the model, the transmission direction of incident light is along the direction of −Z, and the polarization direction is along the *x*-axis. From Figure 6a, it can be observed that a single SiO_2_ nanosphere focuses the incident electromagnetic energy into the solar cell region. As the incident light is transmitted into the device, the electric field gradually diverges, similar to the focusing phenomenon of a single macroscopic lens. As shown in Figure 6b, when two vertically arranged nanospheres are located on the surface of the device, they do not show apparent focusing and divergence effects. We further calculated the electric field distribution on different Z planes, as shown in Figure 6c,d. It is found that the electric field distribution caused by a single nanosphere at a distance of 15 nm from the surface in a silicon cell is similar to that of the vertically arranged dimmer. By contrast, a single nanosphere causes a more intensive electric field near the junction area. Therefore, the single-layer ~550 nm SiO_2_ nanospheres will concentrate more light energy into the junction region of the device where the extraction efficiency of photo-generated carriers is largest, thereby effectively increasing the output power.

### 3.3. The Influence of Poly-l-Lysine on the Performance of Solar Cells

Another question we want to discuss is the influence of poly-l-lysine on solar cells. Both optical and electrical properties are discussed. The reflection spectra and the *J*-*V* characteristics of the original S3, S3 treated by the poly-l-lysine, and S3 with 10 min SiO_2_ nanosphere coating are shown in Figure 7a,b. Both the reflectance spectrum and *J*-*V* characteristic cure of S3 are overlapped with those of S3 treated by poly-l-lysine. Besides, S3 with 10 min SiO_2_ nanosphere coating shows a noticeable reduction in reflection and increased current density compared to that of the original S3. That is to say, the influence of the poly-l-lysine could be neglected on the performances of solar cells.

### 3.4. Optical Analysis of Textured Si Solar Cells with SiO_2_ Nanosphere Coatings

In order to analyze the influence of the introduction of SiO_2_ nanocoating on the optical properties of surface-textured Si solar cells, we systematically analyzed the optical properties of the devices.

First, we test the macro-reflectance spectra of uncoated and coated solar cells. As shown in Figure 8a, when there is only one single layer of SiO_2_ coating on the surface of the device (one round with assembly time of 10 s, 30 s, 10 min), the reflectivity of the device is reduced in a broad spectrum. Moreover, the larger the particle distribution density, the more significant the decrease in reflectivity of the device. When the assembly time is 10 min, that is, when the close-packed condition is reached, the reflectance of the device is the lowest. On the contrary, the reflection will increase (especially in the 700 nm to 850 nm region) when two-layer multi-layer nanocoating is applied (also shown in Appendix A). The change in reflection spectra is well-matched with the variation of photocurrent of the device. We noticed no narrow-band peaks in the reflection spectrum (the feature of WGM) in all textured devices. On the textured surfaces, the height of the nanospheres is different, so the WGM effect caused by the planar close-packed photonic crystals no longer exists [28,29,30]. To further analyze the reflectance variation at different wavelengths, we normalized the reflectance of the device with SiO_2_ nanosphere coating to that of the uncoated device. As shown in Figure 8b, there are two dips in the normalized reflectance spectra. The first dip is in the 550–600 nm band, and the second is in the 800 to 850 nm band. Moreover, as the particle distribution density increases, both dips have a redshift. Macroscopically, the equivalent refractive index can be used for analysis. When the particle density increases, the equivalent refractive index of the nanosphere coating increases. Similar to the anti-reflective coating in the device, the increase in refractive index causes a redshift of the reflection spectrum [29].

Then, brightfield and darkfield optical microscopy imaging techniques are used to analyze the influence of nanocoating with different assembly times and rounds on the optical reflection and scattering of the textured solar cells. A wide-spectrum halogen tungsten lamp was used as the white light source. Besides, a scientific-grade color camera is used to collect micro-region brightfield (Figure 9a–e) and dark-field graphs (Figure 9k–o). The color graphs were transformed into grayscale images by a grayscale transformation code (MATLAB), shown in Appendix A. The gray image divides the intensity into 256 levels. We analyzed the number of pixels on different gray levels in the form of histograms, shown in Figure 9f–j, for brightfield images, and Figure 9p–t for darkfield images. In order to analyze the intensity and contrast of the image, we further calculated the average and standard deviation (std. in Table 2) of the intensity of the brightfield grayscale image and the darkfield grayscale image, as shown in Table 2.

Brightfield images are usually used to reflect the specular reflectance of the devices [46,47]. By adding a single-layer SiO_2_ coating, the specular reflectance of the textured solar cell is obviously reduced. When the assembly is carried out for two rounds, the average reflection intensity is increased. Therefore, one-round assembly for 10 min yields the lowest average reflection intensity value. Darkfield images reflect the large-angle scattering ability for the textured solar cells [48]. It can be found that the textured surface of polycrystalline Si solar cell exhibits strong backward light scattering, especially in the sharp regions. By adding SiO_2_ nanospheres, the backscattering of these sharp features can be weakened. At the same time, the SiO_2_ nanospheres could also enhance the scattering of the original silicon devices in areas with extremely weak scattering. Statistics data show that the average backscattering on the device surface was the weakest when 10 min one-round assembly was conducted. Fewer particles or multiple layers of particles will increase the backscattering. The brightfield and darkfield analysis verify that the specular reflection and backscattering of the textured solar cell can be simultaneously suppressed when coated with a single-layer close-packed SiO_2_ nanocoating. In addition, the standard deviations of the brightfield and darkfield scattering intensities are gradually reduced with increased SiO_2_ nanosphere distribution density. The trend still holds after a five-round assembly (Appendix A). That is to say, except for reduced reflection and scattering, SiO_2_ nanocoating also makes reflection and scattering intensity more uniform from the textured Si surface.

## 4. Conclusions

In this work, a simple assembly scheme on a 3D substrate with controllable distribution is demonstrated. By employing this strategy, for the first time, we realize the controllable assembly of SiO_2_ nanospheres on the surface of textured silicon. The distribution density can be varied from sparse to dense, and the number of layers can be varied from single-layer to multi-layer. This assembly method allows us to study the effect of SiO_2_ nanosphere arrangement on the performance parameters of textured Si solar cells. The optimized enhancement was achieved by close-packed single-layer nanospheres. The efficiency increased by 9.35%, the current increased by 6.76%, and the fill factor increased by 1.96%.

In comparison with the planar substrate, it is found that 3D substrate promotes dense packing due to gravitational sedimentation effects. The quantitative mechanism analysis still needs further study, including morphology of 3D substrate, the combined effect of gravitational sedimentation, the attractive and repulsive force between particles, and that of the substrate with particles. The quantitative mechanism analysis still needs further study, including morphology of 3D substrate, the combined effect of gravity, the attractive and repulsive force between particles, and that of the substrate with particles. The resulted enhanced fill factor of textured Si solar cells is evidence of the nanosphere concentration effect. Reduced surface reflection and backward scattering play a vital role in photocurrent enhancement. Morphology engineering of the nanosphere may further increase the concentration effects. This research provides heuristic guidelines for nanofabrication and photon management on complex 3D surfaces at the nanoscale.

## Figures and Tables

**Figure 1 nanomaterials-11-02581-f001:**
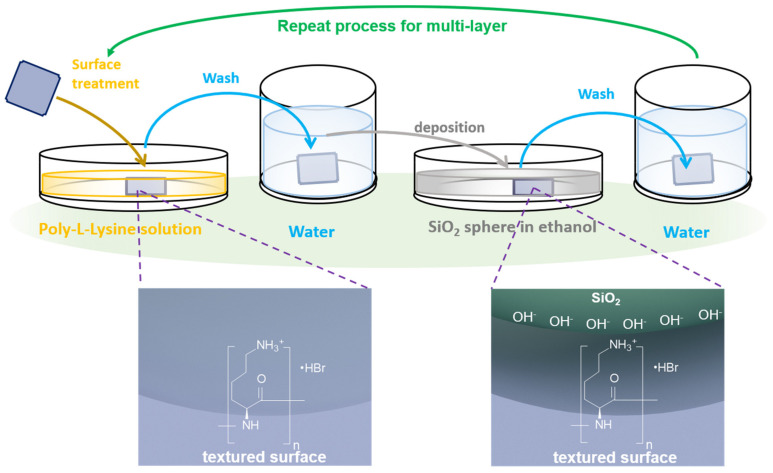
Schematic illustration and self-assembly process. The four-step self-assembly procedure of SiO_2_ nanosphere onto the textured 3D surface, and the inset figures show the electrostatic force-directed assembly mechanism.

**Figure 2 nanomaterials-11-02581-f002:**
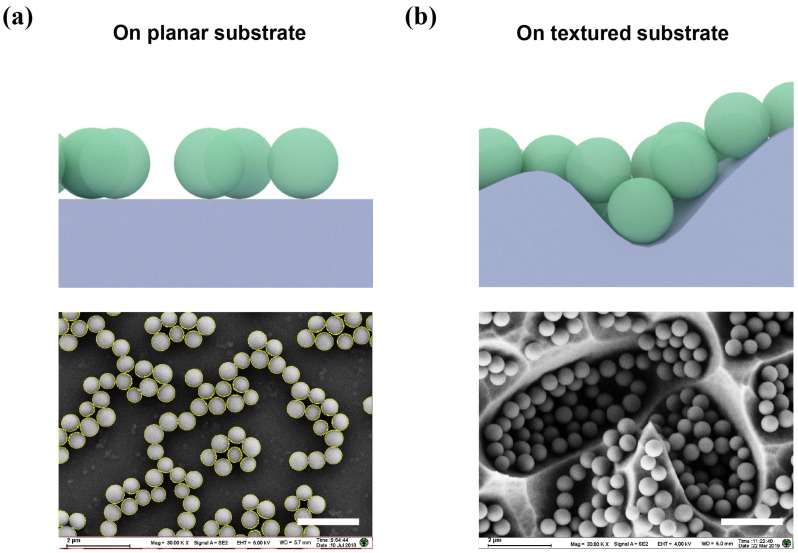
Schematic illustration and SEM graphs of self-assembled SiO_2_ nanocoating on the planar and textured substrates: (**a**) Assembly on a planar substrate, schematic representation (**top**), and the SEM graphs (**down**). The yellow circles around nanospheres were used to calculate the surface coverage of nanospheres with the ImageJ software; (**b**) Assembly on a textured substrate, schematic representation (**top**), and the SEM graphs (**down**). The dipping time in SiO_2_ assembly solution is the same (10 min) for the planar and textured substrate. The scale bar is 2 μm.

**Figure 3 nanomaterials-11-02581-f003:**
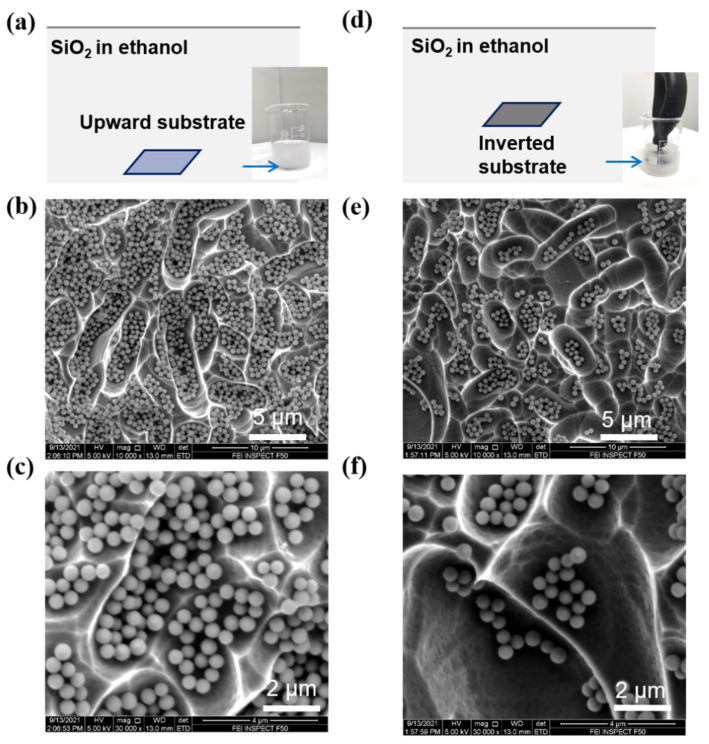
Self-assembly nanocoating under different substrate orientations in the SiO_2_ assembly colloid: (**a**–**c**) Self-assembly with an upward substrate; (**d**–**f**) Self-assembly with an inverted substrate. The assembly time was fixed to 10 min for all the assembly processes, and the inset figures in (**a**) and (**d**) are the images of assembly in colloidal SiO_2_. The blue arrows indicate the location of the devices. A vacuum suction ball was used to keep the substrate upside-down during the assembly process in the inverted condition.

**Figure 4 nanomaterials-11-02581-f004:**
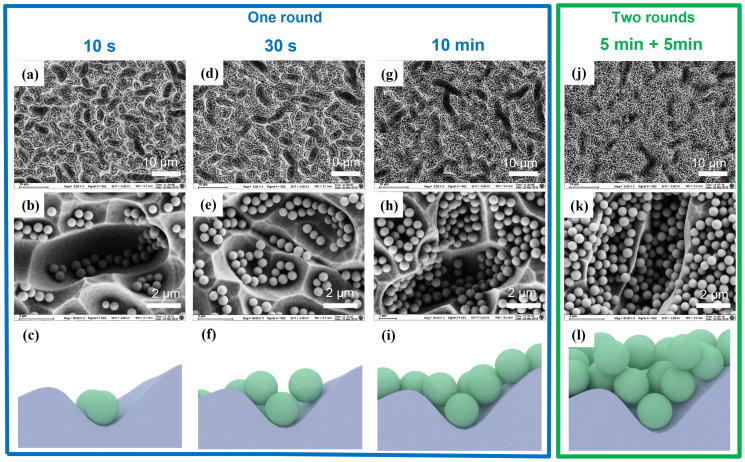
Self-assembly nanocoating under different timelines and repeated assembly process. SEM graphs under different magnification and the schematic diagram: (**a**–**c**) One round with an assembly time of 10 s; (**d**–**f**) One round with an assembly time of 30 s; (**g**–**i**) One round with an assembly time of 10 min; (**j**–**l**) Two rounds with assembly times of 5 min + 5 min.

**Figure 5 nanomaterials-11-02581-f005:**
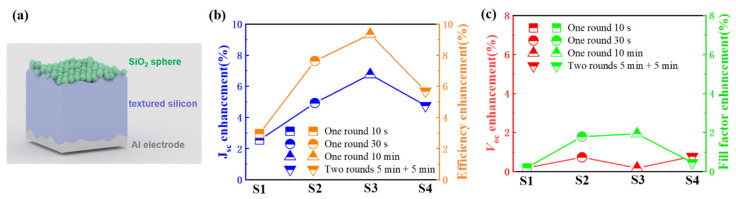
Comparison of electrical parameters enhancement with different SiO_2_ nanocoatings. (**a**) Schematic illustration of the textured Si solar cell with SiO_2_ nanosphere coating; (**b**) Relative enhancement of *J*_sc_ and power conversion efficiency of the textured Si solar cells; (**c**) Relative enhancement of *V*_oc_ and fill factor of the textured Si solar cells.

**Figure 6 nanomaterials-11-02581-f006:**
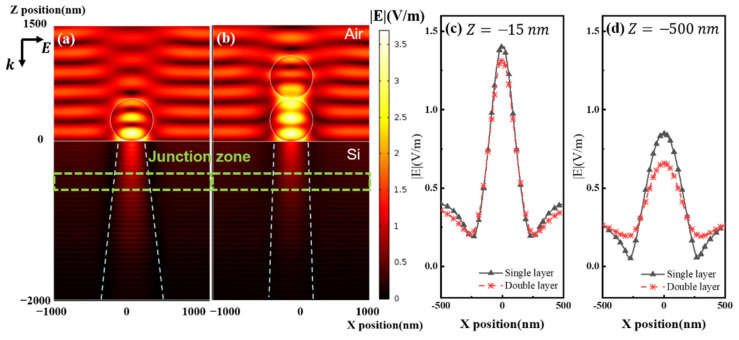
Electric field simulation of single-layer and double-layer nanospheres on Si solar cells: (**a**) Electric field distribution of single SiO_2_ nanosphere atop Si solar cell; (**b**) Electric field distribution of vertically aligned SiO_2_ nanosphere dimmer atop Si solar cell; (**c**) Electric field intensity distribution beneath the Si surface at *Z* = −15 nm; (**d**) Electric field intensity distribution near the junction region of Si solar cell at *Z* = −500 nm.

**Figure 7 nanomaterials-11-02581-f007:**
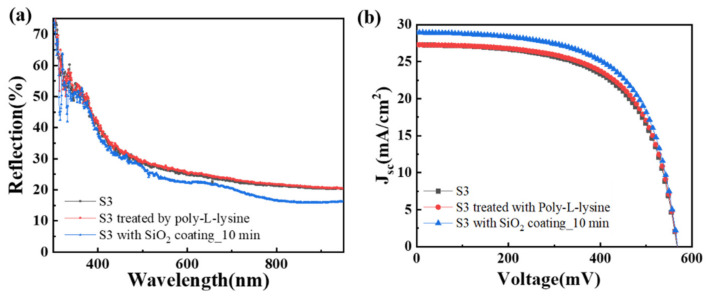
The reflection spectra and current density-voltage characteristics of the solar cells were treated only with poly-l-lysine and after deposition of SiO_2_ nanospheres: (**a**) Reflection spectra and (**b**) Current density-voltage characteristics of the original solar cell (S3, black): S3 after being treated by poly-l-lysine (red), and S3 with 10 min self-assembly of SiO_2_ nanosphere (blue).

**Figure 8 nanomaterials-11-02581-f008:**
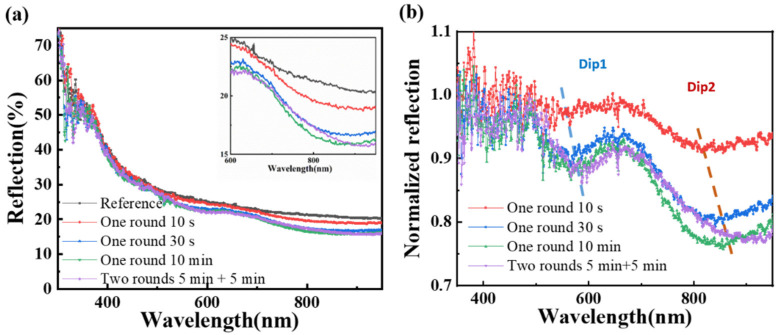
The reflection spectra analysis: (**a**) Reflection spectra from the textured solar cells without or with SiO_2_ nanocoating. The inset graph is the magnified reflection spectra over the wavelength range from 600 nm to 950 nm; (**b**) Normalized reflection spectra of the textured solar cells with SiO_2_ nanocoating.

**Figure 9 nanomaterials-11-02581-f009:**
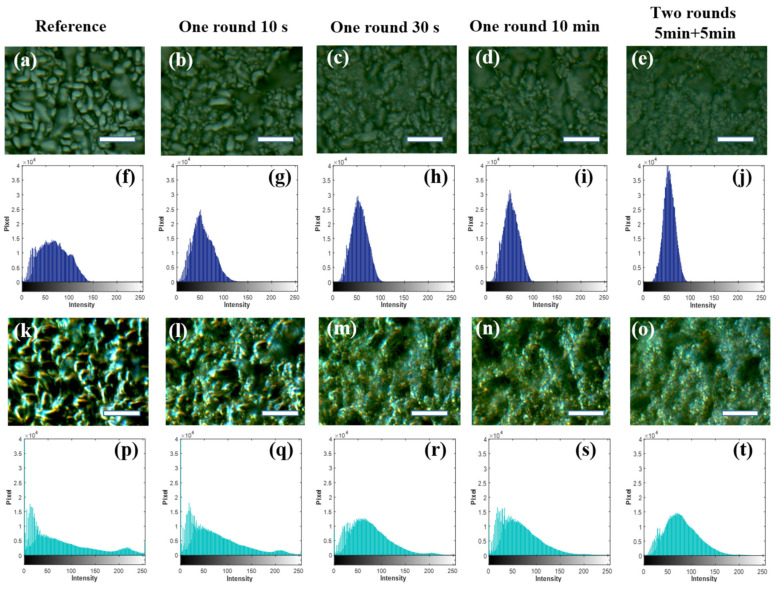
The optical micrographs of the solar cells with different deposition times of SiO_2_ nanosphere coatings: (**a**–**e**) The brightfield images and (**f**–**j**) the corresponding grayscale image histograms; (**k**–**o**) Darkfield images and (**p**–**t**) the corresponding grayscale image histograms. The scale bar is 10 μm.

**Table 1 nanomaterials-11-02581-t001:** Relative enhancement of electrical properties of samples before and after SiO_2_ nanosphere assembly.

Description	*V*_oc_ (V)	*J*_sc_ (mA/cm^2^)	Fill Factor (%)	Efficiency (%)
S1	0.537 ± 0.001	24.98 ± 0.32	50.2 ± 0.1	6.73 ± 0.10
S1–One round 10 s	0.538 ± 0.001	25.62 ± 0.29	50.3 ± 0.2	6.93 ± 0.12
Relative enhancement (%)	0.19	2.55	0.20	2.96
S2	0.546 ± 0.001	24.93 ± 0.13	61.4 ± 0.1	8.36 ± 0.04
S2–One round 30 s	0.550 ± 0.001	26.16 ± 0.17	62.5 ± 0.1	8.99 ± 0.07
Relative enhancement (%)	0.73	4.93	1.79	7.63
S3	0.567 ± 0.001	27.06 ± 0.24	61.3 ± 0.1	9.40 ± 0.10
S3–One round 10 min	0.569 ± 0.001	28.89 ± 0.19	62.5 ± 0.1	10.28 ± 0.09
Relative enhancement (%)	0.18	6.76	1.96	9.35
S4	0.550 ± 0.001	27.13 ± 0.26	64.8 ± 0.2	9.67 ± 0.07
S4–Two rounds 5 min + 5 min	0.553 ± 0.001	28.42 ± 0.26	65.1 ± 0.1	10.22 ± 0.10
Relative enhancement (%)	0.55	4.76	0.46	5.69

**Table 2 nanomaterials-11-02581-t002:** Statistical analysis of brightfield (BF) and darkfield (DF) grayscale images.

Description	Reference	One Round 10 s	One Round 30 s	One Round 10 min	Two Rounds 5 min + 5 min
BF-average	57.76	45.61	46.09	43.59	45.43
DF-average	57.47	61.7	62	53.18	67.48
BF-std.	36.67	29.91	25.92	25.69	23.22
DF-std.	73.42	63.47	51.61	48.54	45.83

## Data Availability

Data are contained within the article.

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
