# Peer review of "Simple Self-Assembly Strategy of Nanospheres on 3D Substrate and Its Application for Enhanced Textured Silicon Solar Cell"

_nanomaterials, 2021, doi:10.3390/nano11102581_

Round 1
Reviewer 1 Report
In the journal article, “Facile Self-Assembly Strategy of Nanospheres on 3D Substrate and its Application for Enhanced Textured Silicon Solar Cell” by Su, Lv, Yang, Zhou, and Zhang, the authors present their results on depositing SiO2 spheres on Si solar cells. In the paper, the authors discuss the antireflection and light focusing effects of nanospheres on 3D textured Si surfaces. The work done in this study is interesting and informative but there are aspects that need to be addressed as listed below:
- The speculation that gravity is a driving force for assembly of the spheres is not well supported by the data and rather dubious. This claim should be further supported by experimental evidence. For instance, if gravity really plays a role, then performing the assembly with the substrate upside-down or at 90 degrees should change the assembly characteristics. I hypothesize that the effects of having the spheres assemble at the bottom of the features is more strongly related to the rinsing step and having the spheres on the pointed parts washed off more easily leaving behind the ones in the trenches. The authors state that investigating the role of gravity is a point for future study (lines 377-379), however, the data do not fully support the gravity-induced assembly mechanism claimed herein and must be addressed for this publication.
- The number of significant digits given for the numbers in Table 1 is unrealistic and should be re-evaluated.
- Lastly, the English grammar and language need improvement. There are many confusing/incorrect/ambiguous sentences that need adjustment. For instance, the sentences on:
- Lines 20-22 is a confusing and ambiguous sentence
- Lines 26-28 eg. “nanosphere-induced concentration and anti-reflection” needs a subject, I would suggest adding “of incident light” to the end of the sentence.
- Lines 39-40 eg. “three-dimensional dimensions”
- Line 50 “focused ionic beam” should be “focused ion beam”
- Lines 52-55 is a long and confusing statement with incorrect grammar.
- Line 129 “long work distance” should be “long working distance”
Reviewer 2 Report
The manuscript describes interesting results on the facile assembling of sub-nanosphere SiO2 on the 3D substrate and its application in a solar cell. The manuscript is well written and well organized. I recommend the manuscript for publication in this journal. But the author is suggested to correct the following:
1. Line 91: cm*2 to cm2
2. Line 103 and 111, PH to pH
Reviewer 3 Report
Title : The word “facile” has connotations of “not challenging” and “simplistic” and should be avoided. What is meant is preumably “simple” which does not have the same connotations. Abstract : Remove “Herein” from “Herein, we”. The use of “linking words” in scientific papers is usually not recommended, as opposed, for example, to their recommended use in literature. Rewrite phase “resulting from the coexisted gravity and the inter-particle forces in the micro-valley” which I not grammatical, the meaning is ill-defined. Rewrite “Meanwhile, distribution density and layers of SiO2 coating could be well”. Avoid “meanwhile” another linking word. Avoud the conditional “could” which is uncertain. Hence omething like “Distribution density and layers of SiO2 coating can be well” Spelling : “tunning” => “tuning”. Conclusion : the language is of good standard, but unfortunately lack of care has left some problems which is unfortunate since it causes delays. Recheck the English language, spelling, and grammar. throughout the paper. Introduction Ungrammatical : “However, the nanostructures usually accumulated and form 56 multi-layer in the valley bottom” While is it acceptable, the style of the introduction is perhaps too focussed on referencing a range of studies with insufficient focus. For example “both top-down and bottom-up self-assembly strategies have been widely investigated, yielding ordered or disordered nanostructures”. “propose a facile assembly” - avoid “facile” as in the title. Comment : The use of “gravity” is not good style. There is mention of This sentence must be rewritten “we analyzed the competition mechanism between the gravity of assembled nanospheres on three-dimensional substrates and the repulsion between particles, and the electrostatic gravity between particles and substrate“ - I see no analysis of competition between weight and electrostatics, only claims in the text. - “electrostatic gravity” is meaningless The discussions of this interplay between “gravity” and electrotatics should in my opinion be removed. Otherwise, much analyis would need to be added without a clear benefit to the paper. 2 materials and methods The phrase “Thus, the reflectivity of those small solar cells was almost the same” is not a consequence of the comments immediately preceding it. There is no justification therefore, no logical flow, in concluding that the samples reflectivity is the same in different devices. The phrase “Thus, the reflectivity of those small solar cells was almost the same” can be removed or a justification must be added. The modelling section is too brief. It is unclear if a commercial modelling package is being used, or if one was written. If the former, the modelling package mut be references. If the latter, the modelling performed must be decribed : the equations solved, and the solution method. If this has previously been published, a reference is sufficient. This is not acceptable as it stands because the reader has no knowledge of what has been done in order to understand the results. 3 Results : Section 3.1 : the discussion of the aggregation mechanisms is not necessary for the paper. Comment : The discussion of the interactions between particles is generalist in tone and does not give confidence that the topic is well understood. I would recommend removing the discussion and simply referring to [44] for an explanation of elf-aggregation mechanisms rather than an imprecise summary which only raises questions of correct interpretation. While this is not wrong, it does not improve the potential impact of the paper. The use of “multiscale” must be justified. At present, there is no justification of multiscale analysis. This echoes the comment on the aggregation mechanisms. It is not helpful to quote analytical methods without concrete work to justify such quotes. A relevant phrase is “we systematically analyzed the multi-scale 300 optical properties of the devices”. The work described consists of analysis of reflectivity in terms of particle distribution. There is no multicale analysis in terms of relating physical properties at different length scales from the atomic to the mesoscopic to the macroscopic length scales to warrant the claims. This should be rewritte to remove “multiscale” claims, or to include detailed multiscale analysis. Conclusions : No comment. Overall comments : This paper suffer from a lack of care and attention on the writing. There is a significant shortfall on the linguistic level due to a lack of care. The paper suffers from a lack of precision and a use of generalised and discussions on several fronts. It reads in several sections as using keyword from the literature which are correctly referenced but which are then quoted with a claim of excessive detail without jutification. One example is the analysis of mechanisms involved in the self-aggregation of nanoparticles. Another example is the claim of multiscale analysis. This paper being essentially experimental work, the references to these methods of analysis should either be removed except for referencing, or developed and included in detail. Therefore although the paper contains good experimental work, it nevertheless requires a rewrite based on the poot language and the analysis which fall short of the claims of analysis in the text.Author Response
Please see the attachment.

Round 2
Reviewer 1 Report
In this round of review the authors have edited their manuscript and show some new experimental results which still do not support their claims. The entire discussion of gravity being an important force for assembling the particles is skewed. The authors show images of the spheres assembled into the textured trenches when the substrate was inverted in the solution and the spheres are still drawn to the bottom of the textures meaning that the assembling force is not gravitational. The fact that there are more spheres assembled when the substrate is upright means that the only role gravity plays is that the spheres precipitate from solution so there are just more spheres gathering up on the substrate but not that gravity induces their assembly. A rough calculation of the gravitation force between particles yields similar magnitudes to the buoyant force on the particles which indicates that these effects are more than likely negligible. The discussion about gravity assembling the particles should be removed before publication as it is not entirely correct nor complete and is also not needed to explain the other results in the paper.
Reviewer 3 Report
The revision is insufficient. See attached file.
